# Assessing Health Vulnerabilities and Adaptation to Climate Change: A Review of International Progress

**DOI:** 10.3390/ijerph15122626

**Published:** 2018-11-23

**Authors:** Peter Berry, Paddy M. Enright, Joy Shumake-Guillemot, Elena Villalobos Prats, Diarmid Campbell-Lendrum

**Affiliations:** 1Faculty of Environment, University of Waterloo, Waterloo, ON N2L 3G1, Canada; pberry@uwaterloo.ca; 2WMO/WHO Climate and Health Office, 1211 Geneva, Switzerland; jshumake-guillemot@wmo.int; 3Climate Change and Health Unit, WHO, 1211 Geneva, Switzerland; villalobose@who.int (E.V.P.); campbelllendrumd@who.int (D.C.-L.)

**Keywords:** climate change and health, vulnerability assessment, adaptive capacity, adaptation, climate resilient health systems

## Abstract

Climate change is increasing risks to human health and to the health systems that seek to protect the safety and well-being of populations. Health authorities require information about current associations between health outcomes and weather or climate, vulnerable populations, projections of future risks and adaptation opportunities in order to reduce exposures, empower individuals to take needed protective actions and build climate-resilient health systems. An increasing number of health authorities from local to national levels seek this information by conducting climate change and health vulnerability and adaptation assessments. While assessments can provide valuable information to plan for climate change impacts, the results of many studies are not helping to build the global evidence-base of knowledge in this area. They are also often not integrated into adaptation decision making, sometimes because the health sector is not involved in climate change policy making processes at the national level. Significant barriers related to data accessibility, a limited number of climate and health models, uncertainty in climate projections, and a lack of funding and expertise, particularly in developing countries, challenge health authority efforts to conduct rigorous assessments and apply the findings. This paper examines the evolution of climate change and health vulnerability and adaptation assessments, including guidance developed for such projects, the number of assessments that have been conducted globally and implementation of the findings to support health adaptation action. Greater capacity building that facilitates assessments from local to national scales will support collaborative efforts to protect health from current climate hazards and future climate change. Health sector officials will benefit from additional resources and partnership opportunities to ensure that evidence about climate change impacts on health is effectively translated into needed actions to build health resilience.

## 1. Introduction

Growing scientific evidence suggests that climate variability and change pose serious risks to people living in developing and developed countries across the globe [1,2,3,4]. Extraordinary gains to global population health achieved in recent decades are at risk of reversal due to the expected impacts of climate change [2,5]. Countries that lack adaptive capacity, vulnerable populations (e.g., indigenous peoples, people relying on agricultural livelihoods) and certain regions facing severe challenges, such as small island developing states and the Arctic, are more vulnerable to its impacts [4,6]. The efforts of countries to achieve key sustainable development goals (SDGs) under the 2030 Agenda for Sustainable Development are threatened by climate change (e.g., SDG 1—No poverty; SDG 2—End hunger; SDG 3—Good health and well-being; and SDG 6—Clean water and sanitation) [7]. For example, climate change could result in 100 million more people living in extreme poverty [4]. 

Health authorities and researchers, from international to local levels, recognize the serious threats to health that climate change poses and are taking protective measures to reduce current impacts and future risks [8]. Evidence-based information about current and possible future risks to health, vulnerable populations, and effective adaptation options is needed so as to prepare individuals and communities for the health impacts of climate change [9,10]. This includes identification of innovative adaptations for use by public health officials to understand and respond to more severe and possibly compounding effects of future climate change, such as threats from tipping points and shock events that are outside of the range of current experience [11]. 

Climate change and health vulnerability and adaptation assessments (V&As) are an important instrument and process to establish partnerships and obtain information for understanding and addressing climate change-related risks [8]. They can also provide the knowledge needed to realize potentially large health co-benefits from well-designed adaptation and greenhouse gas mitigation measures, thereby taking advantage of what the Lancet Commission on Climate and Health suggests is the “…greatest global health opportunity of the 21st century” [5] (p. 1).

This paper reviews the evolution of V&As as part of efforts by governments and experts to better understand and respond to climate change risks to health. It provides information on the type of guidance that has been developed for conducting V&As and summarizes national and subnational assessments that have been conducted. The importance of assessments in helping to build the global evidence-base of climate change impacts on health is highlighted along with observations on the implementation of findings to support health adaptation action. Recommendations are then provided about supporting future V&As, establishing local to global baselines of climate change health risks, vulnerabilities and adaptation activities and applying the knowledge generated to protect the health of populations. 

## 2. Methods

To estimate the number of national V&As that have been conducted expert knowledge and the records of the World Health Organization (WHO) were used. WHO sourced data consisted of the results from the 2015 WHO Climate and Health Country Survey reported through the WHO/United Nations Framework Convention on Climate Change (UNFCCC) Climate and Health Country Profiles [12], the unpublished results of a WHO commissioned review of assessments, and different reports from the WHO Regional Offices available online. The WHO Climate and Health Country Survey collects global data on a suite of indicators measuring global progress on climate change and health every two years. These indicators include the number of countries that have conducted a V&A.

An additional semi-structured search of the Web of Science database revealed 190 articles with possible discussion of completed national-level climate change and health V&As. Of these 190 articles, 110 underwent abstract review, and 39 received a full-text review for references to national-level studies. The review results supported the WHO data. A review of the United States Agency for International Development (USAID) database revealed a further 3 V&As that have been conducted—in Uganda, Senegal and Mozambique—with the assistance of USAID’s African and Latin American Resilience to Climate Change (ARCC) Program.

## 3. Background

### 3.1. Climate Change and Health

The global evidence base of information about climate change impacts on health has grown over the last few decades through the publication of Intergovernmental Panel on Climate Change (IPCC) reports [2,13], analysis of the global burden of disease from climate change [1,14] and related studies [5,6,15]. Direct and indirect health impacts associated with climate change are caused by rising temperatures, altered precipitation patterns as well as increasingly severe and frequent extreme weather events [2,4,5,16]. Direct health impacts arise from hazards such as heat-waves, droughts and storms, and indirect impacts come from exposures to disease vectors and air and water pollution. Rising carbon dioxide levels, which contribute to climate change, may also reduce the nutrient value in staple crops. This could increase food insecurity among some populations, particularly those in developing countries [17]. A range of social factors can act to either exacerbate the health impacts of the environmental effects of climate change or to help mitigate them with public health interventions (Figure 1). Knowledge gaps about the impacts of climate change on public health, food distribution, poverty, rural communities and indigenous groups and marginalized people exist [4].

Many of the health impacts from climate variability and change can be lessened or avoided through well-designed adaptation measures [2,8]. Health adaptation refers to “the process of designing, implementing, monitoring, and evaluating strategies, policies, and measures intended to reduce climate change-related impacts and to take advantage of opportunities” [19] (p. 501). Public health authorities have decades of experience with known, effective interventions for reducing many health risks from climate and weather-related hazards, such as air and water pollution, contaminated food, vector-borne diseases, ozone depletion, and extreme weather events (e.g., heatwaves, floods, droughts, wildfires, ice storms, hurricanes) [9,20].

### 3.2. Climate Change and Health Vulnerability and Adaptation Assessments

Climate change and health vulnerability and adaptation assessments are conducted at local, regional, or national scales by health authorities to identify and interpret information needed to prepare health systems for the impacts of climate change. These studies are participatory in nature, engaging a range of stakeholders to support efforts in preparing for climate change. They possess a common set of goals and processes, although individual studies may vary significantly in design. Key functions of assessments include [9]:Improving evidence and understanding of the current associations between weather/climate and health outcomes, including the health of populations most vulnerable to these risks;Providing health and emergency management officials, stakeholders, and the public with information on the magnitude and pattern of current and future health risks associated with climate variability and change, and identifying vulnerabilities in the health system;Identifying opportunities to incorporate climate change concerns into existing policies and programs designed to manage health risks associated with weather and climate, and to develop new programs where necessary to prevent and reduce the severity of future risks;Serving as a baseline analysis against which future changes in risks and in associated policies and programs can be monitored;Forging collaborations with sectors such as water and infrastructure to promote activities to improve population health in a changing climate; andStrengthening the case for investment in health protection.

The ultimate goal of an assessment is to support the implementation of adaptation and risk management measures that are effective in reducing current and future climate-related risks to human health and well-being, including among populations most vulnerable to the impacts. V&As develop evidence-based adaptation plans or strategies by health authorities in collaboration with a broad range of partners within and outside the health sector [9]. For least developed countries and other developing countries, Health National Adaptation Plans (HNAPs), and the Operational Framework for Building Climate Resilient Health Systems (2015) are complementary tools designed by WHO to apply information from a V&A to define strategic goals and plans for building health resilience to climate change. HNAPs, as sector specific adaptation plans, can be included as the health component of a country’s National Adaptation Plan (NAP) and highlight national health adaptation goals to be achieved over a specific timeframe, and with available resources [21]. The NAP process was created by the UNFCCC climate change agenda, and facilitates efforts by these countries to undertake medium- and long-term adaptation planning [22].

## 4. Results

### 4.1. Guidance for Conducting V&As 

The WHO, World Meteorological Organization (WMO), and the United Nations Environment Programme (UNEP) established a task team in 1995 to assess risks to health from climate change, setting the stage for two decades of further climate and health assessments [23]. McMichael et al. [23] laid the foundation for the evidence-base on climate change and health by describing what is known about the basic biology and epidemiology of diseases that were first characterized as “climate sensitive”, the direct and indirect pathways through which current climate variability and future climate change can impact health. They also described key decision-making processes supporting health adaptation [23]. In addition, analysis of health impacts of climate change has been included in all five IPCC reports since 1990 and most recently, in the IPCC Special Report on Global Warming of 1.5 °C [4]. In 2014, WHO published quantitative estimates of climate change effects on global health [1].

Effective measures to protect health from climate change impacts require assessments that further the understanding and integration of information about local and population level vulnerabilities [24]. This includes analysis of exposure and sensitivities to hazards, regional weather and climate drivers; such studies benefit from participation of a range of stakeholders at the community level [9,25]. To facilitate efforts by public health officials to assess current and future impacts of climate change on health, guidance and methods for conducting assessments have been developed over the last two decades by building on fundamental principles and theoretical models from environmental health impact assessment [9,14,26,27,28,29,30,31]. The first comprehensive guide for conducting climate change and health assessments was developed by the WHO European Office, in collaboration with WMO, Health Canada (HC) and UNEP in 2003 [27]. It presented a framework and key steps for conducting an assessment, and described quantitative and qualitative methods to establish associations of local climate and weather conditions with health outcomes using specific health impact areas as examples [27]. The guidance also included discussion of useful tools and methods for conducting assessments based upon the principles of environmental health impact assessment, including environmental epidemiology (e.g., short and long time-series studies, ecological studies), geographical information systems, climate change, population and socioeconomic scenarios, climate and health modelling, biological models of infectious disease transmission, development of indicators, comparative risk assessment, integrated assessment, traditional knowledge, literature reviews, single event case studies, focus groups, participatory workshops, and cost-benefit analysis [27]. The report was generally tailored for researchers and provided limited information on health policy and program development. The assessment guidelines were reported to WHO to be complicated and cumbersome by a number of health authorities using them. 

In 2008, the Member States of WHO requested development of tools for conducting V&As through a World Health Assembly Resolution [32]. A new WHO workplan on climate change and health in 2009 included a strong emphasis on building evidence through such studies [33]. WHO therefore developed new guidance which provided stronger direction on the assessment process including how to engage stakeholders and communicate the assessment findings. It also provided greater detail on the adaptation assessment process, such as identifying resources for implementation of options, potential barriers to action and inaction to protect health, as well as their approximate costs [9]. The guidance was developed and tested by WHO and its Regional Offices over 2007–2010. The document included 25 examples of assessment step implementation from a wide range of countries to make the guidance more accessible to public health officials in both developed and developing countries. A survey was administered in 2010 on the experiences of 12 countries that conducted V&As using the new guidance, and a workshop was held in Costa Rica which convened health officials and experts from 15 countries to provide input toward development of the new document [34]. The new guide “Protecting health from climate change: Vulnerability and adaptation assessment” [9] was published by the Pan American Health Organization (PAHO) and WHO and outlines 5 steps (Table 1) for conducting an assessment based upon the 2003 guide. It remains the current tool for undertaking such studies promoted today by the WHO. 

Guidance for conducting V&As has evolved over the last two decades to better accommodate the decision-making needs of public health officials in both developing and developed countries. Changes made reflect needs for more thematic and localized information and data to assess risks over multiple timescales, and to inform adaptation of specific health programs or infrastructures. For example, the Ministry of Health and Long-term Care in Ontario developed a Climate Change and Health Toolkit [29] to assist the 36 local boards of health in that province meet the requirement in the new provincial public health standards to assess the health impacts of climate change [35]. The Toolkit includes technical assessment guidelines that provide case studies of health adaptation and multisectoral collaboration in Ontario, a vulnerabilities assessment checklist, and an assessment workbook with fillable worksheets that can be used to obtain and analyze needed data and information [29].

To cater to regional needs, the WHO, WMO, HC, and PAHO developed guidance to conduct assessments in small-island developing states in the Caribbean with more explicit and regionally relevant direction, including appropriate regional resources, data, and indicators [30]. In response to the request from countries for more detailed guidance for key priority climate change and health issues, WHO is developing supplementary tools for the 2013 guide [9] including information on assessing climate change impacts on zoonotic and vector-borne diseases, diarrheal diseases, airborne and respiratory diseases, heat stress, undernutrition, natural disasters and extreme weather events, and occupational health. Detailed national and thematic guides have also been developed in Canada, the United States, the United Kingdom, and by PAHO/WHO for assessing specific components of health systems such as climate-related vulnerabilities and adaptation options of health facilities [36,37,38,39]. The World Bank (WB) in 2017 developed tailored V&A guidance to inform their investment decision-needs in the health sector [31].

Climate change and health adaptation and vulnerability assessments are most helpful in planning for future risks to populations when they take into consideration future demographic, socioeconomic and health sector trends that may combine with climate drivers to either increase or reduce the resilience of health systems. The Shared Socioeconomic Pathways (SSPs) which were developed to help understand challenges to climate change adaptation and greenhouse gas mitigation activities under different futures are a useful tool for V&As [40]. 

New and innovative approaches to conducting V&As have also come from individual countries. The UK Climate Change Risk Assessment developed a new way to conduct risk framing and prioritize adaptation options to generate more decision-relevant information. This approach was applied to nearly sixty individual risk areas including human health [41]. It used the concept of urgency to summarize the findings of the assessment analysis and to communicate the need for further adaptation actions based upon the four categories: “more action needed”, “research priority”, “sustain current action”, or “watching brief” [41].

### 4.2. Overview of International, National and Subnational Assessments 

Climate change and health vulnerability and adaptation assessments have been conducted at national and subnational scales for over 20 years. The mandate and resources for conducting V&As among countries varies greatly. While many assessments in developing countries have been supported by WHO as part of its technical support to countries, others have been triggered by the need to provide sectoral inputs to National Communications to the UNFCCC, and in some instances, are government mandated. For example, national climate change assessments are mandated in the U.S. by Congress through the Global Change Research Act (GCRA) of 1990 and assessments in the UK through the 2008 Climate Change Act. The most recent US assessment report was produced by a team of more than 300 experts guided by a 60-member Federal Advisory Committee [42]. The UK Climate Change Risk Assessment (comprehensive of all sectors) took 3 years to complete and included commissioning of original research [43]. In contrast, V&As completed by many developing countries may involve only 12–30 national experts engaged for shorter time frames (e.g., three to nine months) and those supported by WHO in the past had much smaller budgets, on average between 50,000–100,000 USD.

A majority of national climate change and health assessments in developing countries have been supported by the WHO (approximately 48) as part of its role in providing technical support to countries for the implementation of climate change and health adaptation projects [34,44,45,46,47,48]. Other agencies such as United States Agency for International Development (USAID) (e.g., Senegal, Mozambique) [49,50], the Asian Development Bank (e.g., Cambodia), the German Corporation for International Development (GIZ) (e.g., Grenada) have also invested to support V&As to inform health adaptation programming. The WB conducted climate change and health assessments in Madagascar and Fiji to ensure that their health investments in those countries were resilient to climate change [31,51].

The number of V&As has risen steadily in recent years in both developed and developing countries, in response to growing investment in climate change adaptation and the need for relevant information. Kovats et al. [52] identified 16 countries as having completed at least 1 national level assessment prior to mid-2002 with half (8/16) having been completed by UNFCCC Annex 1 (e.g., developed) countries. The only countries that undertook multiple assessments prior to mid-2002 were both Annex 1 counties (Australia and Japan) [52]. Only recently have mechanisms been established to systematically and regularly monitor and track such studies. In total, 92 countries have completed national V&As (Figure 2) (Appendix A). Some countries have also conducted subnational assessments (e.g., Ghana, Bangladesh and Nepal) and further inquiry would be needed to obtain a comprehensive estimate of the total number of these studies. 

Analysis was conducted on a sample of the WHO supported V&As completed since 2013, including 34 at the country level and 5 at the subnational level [53]. The 34 assessments were supported by WHO and results of the studies were used to develop WHO Climate and Health Country Profiles. The analysis revealed that all but 4 of 34 assessments included some analysis of the climate change and health vulnerability and/or adaptation capacity [53]. However, for each priority health issue only about one third of the assessments identified specific populations vulnerable to the impacts, leaving decision makers with limited information to develop needed targeted public health interventions. In addition, approximately half of the assessments included either qualitative or quantitative projections of future health impacts of climate change.

Regarding health concerns of focus, all of the 34 assessments identified vector-borne diseases as a priority, while only 16 identified heat and cold-related health risks. The most commonly prioritized individual diseases among vector-borne diseases were malaria (*n* = 28), dengue (*n* = 18), and leishmaniasis (*n* = 8) [53]. Just over half of the assessments (*n* = 20) included projections of future health impacts from climate change, either through quantitative or qualitative analysis, suggesting that many reports included little information about future risks from climate change. Of the 34 studies reviewed, 26 assessments indicated plans or intent to use the study results to inform national climate change adaptation strategies or to enhance the resilience health programming [53]. It is not known whether these findings apply to the rest of V&As conducted to date with or without support from WHO.

Subnational assessments allow for more localized analysis of the health impacts of climate change including a more accurate estimation of current and future vulnerabilities to inform effective adaptation planning. They may also allow researchers to focus limited resources on priority areas (e.g., large urban centers, areas with high proportions of vulnerable people, areas believed to be at increased risk, such as coastal areas). In geographically large countries, such as Canada and the United States that span multiple ecozones/biomes, the anticipated health impacts of climate change may vary widely from one area to another [15,54]. Subnational assessments provide the opportunity to identify and address locally specific health risks that may otherwise be overlooked in a national context (e.g., an invasive vector-borne disease only likely to impact a particular region of a country) and develop locally appropriate response options.

Since 2009, the United States Centers for Disease Control and Prevention (US CDC) has supported state, tribal, local, and territorial public health agencies to use evidence-based information to better understand and predict the health risks of climate change and prepare programs to respond and protect health. Through the Climate-Ready States and Cities Initiative (CRSCI) and the supporting Building Resilience Against Climate Effects (BRACE) Framework [55] it has provided new assessment tools and opportunities for information exchange (e.g., communities of practice) to support 16 states and 2 cities to conduct V&As and develop adaptation plans or measures. A review of CRSCI activities found that the program had increased the capacity of local health authorities to prepare for and respond to climate change by providing information about climate change impacts on health and vulnerable populations [56]. Specifically, the program supported development of practice-informed guidance on quantitative risk assessment methods, greater integration of health considerations within the climate change agenda in the respective jurisdictions and the creation of new networks for sharing information. Experiences of grantees revealed the need for greater efforts to enhance knowledge of climate modeling tools and their use in disease burden projections among public health professionals [56]. Examples of US states and cities that have developed climate change adaptation plans or frameworks or taken other actions based on the research results are presented in Table 2.

Subnational assessments have been completed in a number of other countries as well, including, for example by the Simcoe Muskoka District Health Unit in Canada [57], and the Government of Western Australia [58]. 

## 5. Discussion

Irrespective of greenhouse gas emissions reductions, climate change will continue for decades resulting in gradual warming punctuated by stresses and shocks that have the potential to severely impact individuals, communities and health systems. Learnings from engagement with researchers and decision makers in V&As suggest that climate change is transforming the milieu in which public health officials and partners understand health risks and make decisions to reduce them due to:Dynamic and complex disease and health impact risks which occur over multiple timescales and geographies (e.g., vector-borne diseases, mental health);Multiple uncertainties—particularly around management of cascading and indirect health effects (e.g., food insecurity and nutrition);An increased probability of extreme and “surprise” events that can severely impact health outcomes and health system functioning; andRisks of “involuntary” adaptation (e.g., rushed and poorly planned interventions in response to an emergency or disaster).

Climate change and health adaptation and vulnerability assessments can provide the evidence-based information to help public health officials and partners in other sectors navigate these challenges and respond effectively to increasing risks.

### 5.1. Assessment Opportunities

Climate change and health vulnerability and adaptation assessments have assisted some health authorities and researchers in efforts to raise awareness of possible impacts on vulnerable populations, build capacity to prepare for impacts, understand data and evidentiary requirements for decision-making, and establish the required knowledge base for climate change adaptation in the health sector [9,23]. Assessments have also helped health authorities to integrate the needs of the health sector into national climate assessment and policy efforts. Ministries of Health, which have undertaken V&As, are better prepared to join national and international policy fora and related processes on climate change (e.g., national adaptation planning and strategy development, UNFCCC negotiations, World Health Assembly), and to advocate for action which benefits health and builds the climate resiliency of the health sector.

The horizontal and participatory nature of V&As also facilitates climate change cooperation across sectors. Involvement of professionals from water, energy, conservation, transportation, urban planning, agriculture, coastal management and related sectors provides information about health risks associated with key drivers from impacts or adaptation actions in these sectors [26]. This collaboration opens door to identifying adaptation and risk management responses across sectors that are often preventative in nature, more effective, cost less to society and maximize health co-benefits.

As V&As by health authorities generate more robust evidence, this information can be integrated into international scientific reports including regular studies by the IPCC, the Lancet Commission on Health and Climate, and WHO, thereby providing a more comprehensive global evidence-base of the vulnerability of people to climate change impacts and progress being made to reduce risks. International reports facilitate a common understanding of the risks posed by climate change, improve access to climate and health data, support calls for action to address climate change and inform the development of new tools and methods for assessment. The WHO V&A assessment guide [9] was informed by leading edge concepts, methods and frameworks related to climate impact assessment developed by bodies, such as the IPCC. In turn, evidence regarding risks and vulnerabilities to health from current climate variability and future climate change presented in these international reports contributes to the development of local to national V&As.

### 5.2. Assessment Barriers

In the face of increasing calls for more information about climate change impacts on health and interest from health authorities in undertaking assessments, a number of important barriers exist in completing these studies. A key challenge for health authorities conducting V&As is to provide information in timescales relevant for decision making. To be most useful V&As should provide rigorous findings that help policy makers identify both short-term and longer-term adaptation options. However, as highlighted above, WHO analysis of a sample of completed assessments indicated that only half included either qualitative or quantitative projections of future health impacts of climate change. Limited data, a paucity of climate and health models, uncertainty related to climate projections, and a lack of resources, expertise or time often make estimating future health impacts of climate change in a region or community difficult. Robust indicators that can be used to track: (1) vulnerability, risk, and exposure of populations and health systems to climate change hazards; (2) actual impacts from such hazards; and (3) progress in adapting and building resilience are lacking from local to national scales [59]. To ensure consistency in approaches and comparability of findings V&As often consider and apply findings from existing national climate assessments conducted by the government or other sectors, including the same climate projections (e.g., global and regional SRES, RCP projections). This is useful to raise awareness that current health vulnerabilities, if not addressed now, may increase significantly in the future. However, in resource poor contexts this information alone is rarely sufficient to justify and motivate investments in health adaptation.

In the short-term, policy and program managers often require information from V&As to inform near-term decisions related to staffing, procurement, and programming, for example for efforts to address a particular intervention (e.g., heat-health warning systems, health promotion campaign, disaster management plan, food safety program) found to be ineffective. This can present a fundamental challenge in applying the results of V&A assessments to decision-making. Adaptation options identified in assessments may include important recommendations to strengthen health system management, such as strengthened surveillance, training and capacity building, but lack recommendations for immediate actions. In such cases, gaining buy-in of decision makers that action is required is difficult because results pertaining to near-term adaptations are not included in the study. This type of information requires analysis of how current climate and extreme weather events are driving and influencing health outcomes and health system performance in a country or region. This analysis is often not possible with existing resources (e.g., data, financial, expertise, time) for the assessment. However, improved understanding and analysis of current seasonality, inter-annual climate patterns, conditions and projections of extreme rainfall, temperature, and drought in the country can provide decision-relevant information that is easily applied to current planning cycles, and is also likely representative of future climate conditions.

Many V&As also focus narrowly on building the evidence-base of climate-sensitivity related to specific health outcomes of concern for a specific jurisdiction, frequently measured as disease burdens. They often do not investigate in great enough detail the climate-sensitivity of broader components of the health system (e.g., health facilities, health workforce, new adaptation technologies, adaptation networks and partnerships) as identified in the WHO Operational Framework for Building Climate Resilient Health Systems [10]. This focus on disease outcomes can result in insufficient attention to potentially critical vulnerabilities and impacts from climate change on health-related sectors such as a projected collapse of food stocks, large-scale landscape changes due to sea-level rise or hydrological changes, or population migration from climate impacts. It also results in inadequate attention to measuring impacts on the health sector itself, such as those on health systems and facilities and on levels of adaptive capacity; information on such impacts could generate specific actionable recommendations. Future studies would benefit from greater focus on the potential impacts of climate change, including stresses and shocks, on health systems [11].

Particular obstacles may face developing countries in conducting V&As. National analyses of health risks and future impacts are often challenged by the availability of national and local data, which are often insufficient to establish robust linear and non-linear relationships among health outcomes and climate variables. In addition, those conducting assessments often lack both high quality and long-term climate and epidemiological data and have few pre-existing local studies to consider within a V&A. They may also have few climate and health models for application, low capacity within country teams to conduct the analysis, and limited financial resources and time to contract appropriate academic teams to conduct more sophisticated studies. Schnitter et al. [60] reported that challenges encountered in completing a V&A in Dominica in 2015 included insufficient resources and expertise to undertake modeling of future health impacts for more than one issue (vector-borne diseases), limited health data for analysis, and challenges prioritizing adaptation options. 

Robust V&As build upon existing locally relevant evidence, which has already established associations between climate variables, tested data sets, and can describe local vulnerabilities and disease dynamics. However, many developing countries do not have this evidence base, or sufficient data and capacity to conduct this research, and are thus dependent upon proxy studies which may be relevant, but conducted in other locations. This is reflected in WHO’s 2018 finding [54] that only half of the reviewed assessments were able to include future risk projections. V&As in developing countries often focus their resources on establishing associations between weather and health outcomes and not future climate change due to this lack of existing evidence. They often conclude with recommendations for additional studies; closer collaboration between the international research community and local policy communities would help address these gaps.

Challenges associated with mobilizing communities and health authorities to take action on climate-related health risks based upon V&A studies often relate to limited funding for climate change and health activities, particularly in developing countries. The World Health Organization estimated that in 2017, less than 1.5% of available funds dedicated to climate change adaptation globally are used for health protection projects [61]. Limited support for climate change and health and related assessment activities inevitably reduces the number and quality of V&As, making it harder for health authorities to engage in broader climate change activities and build needed capacity. Greater funding would significantly strengthen capacity at all levels of government and among civil society organizations to increase understanding of the health impacts of climate variability and change [25]. Building such capacity would also be supported by expanded training of health officials from local to national levels on conducting V&As with stakeholders and research partners. 

Health authorities often have difficulty effectively translating the findings from a V&A into adaptation practice to protect populations from current climate hazards and future climate change [50]. This includes little application of the knowledge produced into relevant policy processes at the national level such as the NAPs, the UNFCCC National Communications and Nationally Determined Contributions. Challenges can be faced by the health sector in efforts to engage in national climate change processes, which are most often led by the environmental sector. This may also occur because some assessments are not producing information and data most needed by decision makers to protect health and because of a lack of evidence informed activities and tools to support knowledge translation (e.g., spatial tools such as GIS, training kits, education and outreach materials, communities of practice) and facilitate efforts to reduce population health risks. As noted above, while two thirds of the V&As reviewed by WHO indicated intent to use the findings to inform health adaptation planning, only one third identified specific populations vulnerable to climate change health impacts. In addition, Hayes and Poland [62] suggest that many V&As have not included investigation of mental health impacts of climate change, a growing concern among health authorities. Decision makers are therefore often not receiving needed evidence-based information to develop targeted public health interventions to prepare for climate change. 

National V&As have been conducted relatively recently and time is needed to gauge accurately the degree to which findings and recommendations will inform concrete efforts to prepare for climate change impacts on health. It is possible that information from national level assessments is being used by health decision makers at other levels of government (e.g., local, regional) or by non-governmental partners to adapt. This could include efforts to raise awareness of impacts, build networks for cooperation, or pursue targeted actions such as the development of heat-health warning systems that warn the public of impending dangerous temperatures to support protective behaviors. 

V&As by both developed and developing countries have had limited influence on the evolution of the body of climate change and health knowledge reflected by the scientific literature. Such assessments are rarely published, peer reviewed, or communicated outside government-led projects; they draw upon existing literature, and influence local knowledge but are not directly contributing to the global knowledge base.

### 5.3. The Way Forward in Conducting V&As

Climate change and health vulnerability and adaptation assessments are an important instrument for taking stock of current conditions and vulnerabilities, and should be considered as part of an iterative process to be reviewed and returned to over time [9,63]. Establishing health impact, vulnerability and adaptation baselines through a V&A allows health authorities to monitor progress made, or adaptation failures, in efforts to protect the health of populations from climate change impacts. Most countries have not yet repeated their V&A assessment to assess whether the baseline constructed in the first study is sufficient to monitor the success of adaptation efforts and the health outcomes related to climate change. In the future, WHO support for countries to undertake Climate and Health Country Profiles biennially [12] will help provide this information. 

Climate change and health action by health authorities and partners can be strengthened by the development of indictors from local to national scales that capture trends among populations and health systems related to: (1) vulnerabilities, risks and exposures to climate hazards; (2) impacts from such hazards and (3) adaptation and resilience building. Regular tracking of climate change impacts with such indicators supports iterative risk management where evidence is synthesized through V&As and then disseminated and applied through health adaptation [59]. As part of this approach, V&As should be regularly updated to ensure health authorities are using the latest information about key vulnerabilities and are monitoring progress in addressing climate change impacts on health [63].

Participatory approaches to V&As should include involvement of all relevant stakeholders (e.g., meteorological services, Ministry of Environment, Ministry of Health, community social services), experts (e.g., climate change, epidemiology, statistics), and representatives of populations with unique perspectives, or that are at higher risk (e.g., indigenous populations, children, women, elderly people, mentally or physically disabled people, ethnic minorities, impoverished people) [9,29]. Broad engagement with a range of partners improves the quality of assessments while development of a detailed plan to communicate the findings to key decision makers, stakeholders and the research community through a range of opportunities (e.g., UNFCCC National Communications; publication of findings) ensures that the results are relevant and useful for efforts to protect health [9].

Opportunities also exist to inform decision making with V&A findings by integrating results into relevant initiatives within existing mechanisms and processes (e.g., HNAPs, Health and Environment Strategies Alliance (HESA), Situation Analysis and Needs Assessment (SANA), disaster risk reduction) that support efforts to protect health from climate change impacts. Although an important challenge, V&As will benefit from efforts to increase knowledge of local level climate change risks to health and the needs of vulnerable populations. This can be achieved through the conduct of local and regional V&As, as the US BRACE example discussion above illustrates, or by integrating such information into national studies (e.g., investigation of perceptions, adaptive capacity; use of case studies).

## 6. Conclusions

Greater knowledge of risks to health and well-being posed by climate change and guidance available to health authorities for assessing vulnerabilities has resulted in more local, regional and national studies being undertaken. Climate change and health vulnerability and adaptation assessments provide invaluable information to health authorities to increase the resilience of individuals, communities and health systems to climate change impacts. They also contribute to development of the global evidence-base on climate change and health needed to inform adaptation and greenhouse gas mitigation measures in the future, ensuring that they benefit the well-being of populations.

Expanded efforts through V&As to continue to build this global evidence base and inform the development of adaptations to protect against growing risks from climate hazards require further development of both quantitative and qualitative assessment methods, tools and guidance documents [25]. This would include, for example, prioritizing adaptation options, using qualitative methods to assess future impacts from climate change and developing indicators of health system resilience. Greater skill development in environmental health and epidemiology that includes tools and competencies for understanding health risks in the context of environmental and climate change will support health authorities in undertaking V&As. The rigor of assessments would improve with more robust metrics and approaches for estimating adaptive capacity and the effectiveness of current adaptations, as well as with more climate scenario and modelling information applicable to regional and local scales. Enhanced understanding of systemic, synergistic, cascading or compounding health effects from climate change needs to be included in V&As to prepare for potentially severe, or catastrophic events. The use of V&As as a key tool to support effective adaptation to the health impacts of climate change will benefit from increased collaboration, exchange and learning among health authorities as they continue to undertake these studies in the future.

## Figures and Tables

**Figure 1 ijerph-15-02626-f001:**
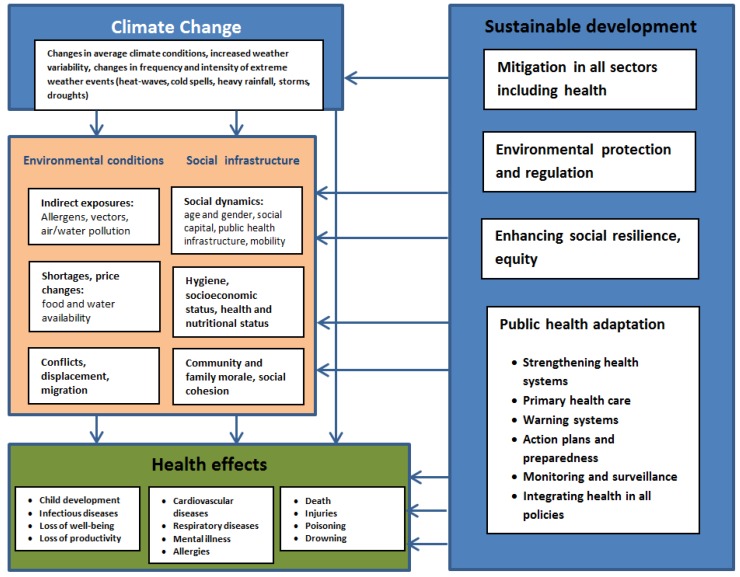
Pathways of climate change, sustainable development and health. Source: [18].

**Figure 2 ijerph-15-02626-f002:**
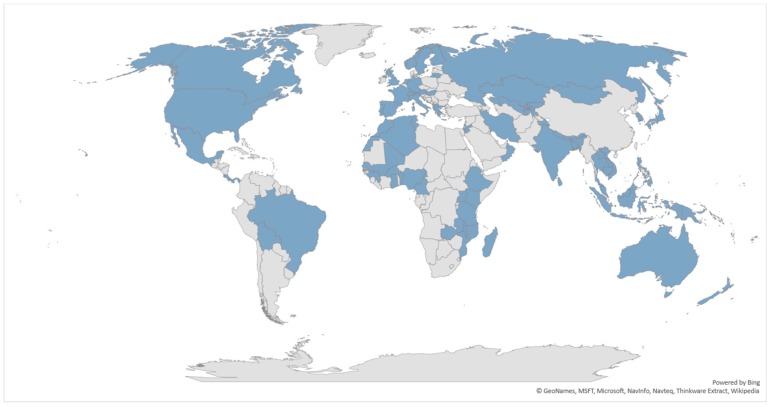
Countries that have completed a national climate change and health vulnerability and adaptation assessment. Not visible on the above map: Antigua and Barbuda, Barbados, Brunei, Cook Islands, Dominica, Federated States of Micronesia, Fiji, Grenada, Kiribati, Maldives, Malta, Marshall Islands, Nauru, Niue, Palau, Samoa, Solomon Islands, St. Lucia, Timor-Leste, Tonga, Tuvalu, and Vanuatu.

**Table 1 ijerph-15-02626-t001:** Five steps for undertaking a climate change and health vulnerability and adaptation assessment. Source: [9].

Assessment Step	Activity
1. Frame and scope the assessment	Define the geographical range and health outcomes of interest Identify the questions to be addressed and steps to be used Identify the policy context for the assessment Establish a project team and a management plan Establish a stakeholder process Develop a communications plan
2. Conducting the vulnerability and adaptation assessment	Establish baseline conditions by describing the human health risks of current climate variability and recent climate change, and the public health policies and programs to address the risks Describe current risks of climate-sensitive health outcomes, including the most vulnerable populations and regions Identify vulnerable populations and regions Describe risk distribution using spatial mapping Analyze the relationships between current and past weather/climate conditions and health outcomes Identify trends in climate change-related exposures Take account of interactions between environmental and socioeconomic determinants of health Describe the current capacity of health and other sectors to manage the risks of climate-sensitive health outcomes
3. Understanding future impacts on health	Estimate future health risks and impacts under climate change Describe how the risks of climate-sensitive health outcomes, including the most vulnerable populations and regions, may change over coming decades, irrespective of climate change Estimate the possible additional burden of adverse health outcomes due to climate change
4. Adaptation to climate change: Prioritizing and implementing health protection	Identify and prioritize policies and programs to address current and projected health risks Identify additional public health and health care policies and programs to prevent likely future health burdens Prioritize public health and health care policies and programs to reduce likely future health burdens Identify resources for implementation and potential barriers to be addressed Estimate the costs of action and of inaction to protect health Identify possible actions to reduce the potential health risks of adaptation and greenhouse gas mitigation policies and programs implemented in other sectors Develop and propose health adaptation plans
5. Establish an iterative process for managing and monitoringthe health risks of climate change	Establish an iterative process for managing and monitoring the health risks of climate change

**Table 2 ijerph-15-02626-t002:** Climate Change and Health Adaptation Plans or Actions Taken by the United States Centers for Disease Control and Prevention (U.S. CDC) Building Resilience against Climate Effects (BRACE) Climate-Ready States and Cities Initiative Participants.

**US CDC BRACE Climate-Ready States and Cities Initiative** **Examples of Climate Change Adaptation Plan/Framework Developed**
Arizona	New York State
California	North Carolina
Michigan	Oregon
Minnesota	Rhode Island
Wisconsin	San Francisco

**Examples of Other Adaptation Actions or Next Steps Taken**
Florida (priority hazard profiles and cases; county adaptation plans)	Maryland (outreach)
Illinois (public health training, heat toolkit)	Massachusetts (climate change preparedness assessment—local boards of public health)
Maine (Syndromic surveillance system for heat, enhanced vector-borne disease monitoring)	New York City (local hazard mitigation plans—extreme heat, resiliency guideline for infrastructure)
Vermont (public health training)

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
