# Peer review of "Assessing Health Vulnerabilities and Adaptation to Climate Change: A Review of International Progress"

_ijerph, 2018, doi:10.3390/ijerph15122626_

Round 1
Reviewer 1 Report
Assessing Health Vulnerabilities and Adaptation to Climate Change: A Review of International Progress
This manuscript provides a narrative review of international activities relating to climate change vulnerability and adaptation assessments (V&A) including providing an introduction to the history and importance of V&A and the available tools, and identifies and discusses some examples of V&A conducted across a range of countries.
Throughout, there’s a lack of clarity on whether the focus is on V&A or implementing the results of a V&A through an Adaptation Plan. The objectives of this paper as identified at the end of section 1 are focused on V&A. The authors need to adapt their objectives and tighten their focus throughout.
Introduction
P1 line 37: The authors refer to low- and high-income countries, developed and developing nations, UNFCCC Annex … and LDCs throughout. They should stick with consistent language.
P2 line 43: Here and throughout, properly introduce acronyms at first use, and then use acronyms consistently thereafter: SDG, V&A, LDCs, USAID
P2 lines 43-45: The authors list specific SDG’s that climate change can impact, but the example provided relates to an SDG not listed. Please consider including SDG #1 “No Poverty” in this list. Similarly, it’s not clear how climate change could threaten climate action – countries could still take action against climate change while suffering from its impacts.
P2 line 52: Tipping points are part of common language around climate change, however, cascading and shock events are less common. The authors should consider either defining these or citing a document that defines these.
Background
P3 line 99: What is a “surprise” event?
P3 line 95-102: This list appears to be a central theme in terms of the decision making that V&A are necessary to support, but where is the evidence to support this list? More specifically, how were these risks identified, and why where these chosen in contrast to other potential risks, such as tipping points.
P4 line 115: This should read identifying, not identify
P4 Line 123: Strengthening, not strengthen.
Results
The presence of a results section implies results of research. This paper does not include an explicit Methods section which identifies how guidance documents were identified (3.1), how international, national and subnational assessments were identified (3.2) or how the authors identified assessment opportunities and barriers as identified in the discussion. The authors provide some description of their methods at the bottom of page 7, which could serve better as part of a methods section. I think if the authors intend to present this section as results (alternately, they could restructure so this is not framed as results), this manuscript could benefit from a more explicit, even if brief, section outlining the methodology.
P4 line 144: evidence-base, not based
P4 lines 132-148: The authors introduce the 1996 work by McMichael et al as laying the groundwork for identifying climate sensitive health impacts. It would be appropriate to refer to some of the more recent works summarizing the literature on climate sensitive health impacts, such as the IPCC reports or more recent work with the WHO.
P5 line 190: Should this read “decision making needs”?
P5 line 204-206: How do the thematic guides described here differ from the V&A guidelines in Canada/PAHO/etc described in the previous paragraph?
P5 lines 204-206: It may be out of scope here, but organizations such as the Climate and Health Alliance in Australia have also been doing work looking at climate change vulnerability and resilience in the healthcare sector.
Table 1: It is not clearly how the detail defined in the Activity column is necessary for the purposes of this manuscript.
P7 lines 232-234: Are there any references to support this statement about resources available for developing country V&A?
P7 line 242, P9 line 299: What is the difference between a V&A and a climate change and health assessment as referred to here.
P8 line 273-282: Is this all of the assessments identified, or just all of the assessments from the sample of WHO V&As and analyzed in the unpublished WHO report in reference 43? It would be valuable to compare these to V&A that didn’t follow the WHO framework – did those other V&A identify similar health risks, how did they compare in terms of future impact assessment or adaptation planning.
P9 line 301: to prepare [for] and respond to…
P9 line 302: Table 2 only refers to US examples (using the BRACE framework), this should be explicit in the text as well, i.e. US states and cities…
Have the authors explored including reference to other subnational estimates, such as those in Australia and Canada, or those supported by UN HABITAT? For example, there are references to relevant examples included in the IJERPH special issue.
Table 2: Needs a title.
P9 lines 293-304: One of the main findings from BRACE was knowledge and the need for linkages to research institutions to support capacity for local modelling of impacts. The authors should consider describing those here as they refer to them as barriers in the discussion.
Discussion
The authors discuss how these V&A are not translating into adaptation practice, however, a major barrier is the lack of evidence informed activities to support adaptation practice. Further, adaptation does not always happen at a national level, it could be that national V&A are leading to adaptation practice at a subnational level which may be more difficult to capture in this type of scoping review.
P9 lines 312-318: What is the benefit of repeating this list from page 3? If this is a key theme, perhaps it could be highlighted in a table, figure or box and referred to in brief in the text.
P10 lines 325-328: This tool seems important in terms of V&A process, it might fit better in the results section.
P10 line 330: C is missing from Climate in the first line of section 4.1
P10 lines 365-369: This point is not highlighted clearly elsewhere. The results discuss the limited resources in developing countries, however, challenges related to data and model availability have not been emphasized previously.
P10 line 369 – p 11 line 372: Ebi et al in this IJERPH special issue provide a discussion of indicator development for modelling and evaluation of climate related hazards that is relevant here and in the later “way forward” section.
P11 lines 392-403: There disconnect between population health and the healthcare sector may act as a barrier to translation of population health research, e.g. into climate sensitive health outcomes and measuring of impacts and producing action plans within health sectors.
P11 lines 420-421: Which WHO report is this referring to?
P12 line 433: There is a missing period before “As”.
P12 lines 446-447: These V&A often do not represent primary research, but rather a synthesis of other published research and so inclusion in the IPCC reports may not necessarily be appropriate.
P12 line 448: I believe this should be section 4.3
P12 line 448: The previous section clearly identifies barriers to V&A and adaptation planning, this section is an opportunity to describe facilitators for completing (and actioning) V&A.
P12 lines 462-463: Should community and other social services be considered as stakeholders here?
P12 line 463: Does “met services” refer to meteorological services?
P12 lines 465-466: Indigenous voices and traditional knowledge are an important resource, but this has not been discussed elsewhere in the manuscript. This list seems more a result than a discussion.
P12 lines 467-468: The previous section of the discussion talks in detail about barriers to local assessment. The authors have not provided suggestions on how to resolve this challenge or overcome this barrier. Further, there is limited reference to local (as in subnational) assessments in previous sections.
Conclusion
P13 lines 504-506: This point about funding for climate change V&A is important, but it does not seem appropriate to introduce this new information in the conclusion. Consider moving this to section 4.2 where barriers are discussed.
References
4: This reference is incomplete.
Author Response
REVIEWER 1
This manuscript provides a narrative review of international activities relating to climate change vulnerability and adaptation assessments (V&A) including providing an introduction to the history and importance of V&A and the available tools, and identifies and discusses some examples of V&A conducted across a range of countries.
Throughout, there’s a lack of clarity on whether the focus is on V&A or implementing the results of a V&A through an Adaptation Plan. The objectives of this paper as identified at the end of section 1 are focused on V&A. The authors need to adapt their objectives and tighten their focus throughout.
The authors thank the reviewer for the insightful comments on the manuscript. It is hoped that revisions based upon these comments have significantly improved the paper.
The Abstract and Introduction have been revised to clarify that the paper has two objectives including a review of the evolution of climate change and health vulnerability and adaptation assessment guidance and conduct of these studies, as well as application of the findings.
Introduction
P1 line 37: The authors refer to low- and high-income countries, developed and developing nations, UNFCCC Annex … and LDCs throughout. They should stick with consistent language.
Change made. The terms developed and developing countries is used throughout the paper.
P2 line 43: Here and throughout, properly introduce acronyms at first use, and then use acronyms consistently thereafter: SDG, V&A, LDCs, USAID
Change made.
P2 lines 43-45: The authors list specific SDG’s that climate change can impact, but the example provided relates to an SDG not listed. Please consider including SDG #1 “No Poverty” in this list. Similarly, it’s not clear how climate change could threaten climate action – countries could still take action against climate change while suffering from its impacts.
Change made. SDG 1 has been added and SDG 13 has been deleted.
P2 line 52: Tipping points are part of common language around climate change, however, cascading and shock events are less common. The authors should consider either defining these or citing a document that defines these.
Change made. The following document has been cited to support these definitions. Ebi, K.L., Berry, P., Boyer, C., Hayes, K., Enright, P.M., Sellers, S., Hess, J.J. (2018). Stress testing the capacity of health systems to manage climate change-related shocks and stresses. Int. J. Environ. Res. Public Health 2018, 15(11), 2370; https://doi.org/10.3390/ijerph15112370 https://www.mdpi.com/1660-4601/15/11/2370
Background
P3 line 99: What is a “surprise” event?
Change made. “Surprised” changed to “shock”.
P3 line 95-102: This list appears to be a central theme in terms of the decision making that V&A are necessary to support, but where is the evidence to support this list? More specifically, how were these risks identified, and why where these chosen in contrast to other potential risks, such as tipping points.
Change made. Much of the evidence regarding how climate change has and is altering public health decision making derives from the experiences conducting a number of V&As and the associated engagement with decision makers. The text has been revised to reflect this.
P4 line 115: This should read identifying, not identify
Change made.
P4 Line 123: Strengthening, not strengthen.
Change made.
Results
The presence of a results section implies results of research. This paper does not include an explicit Methods section which identifies how guidance documents were identified (3.1), how international, national and subnational assessments were identified (3.2) or how the authors identified assessment opportunities and barriers as identified in the discussion. The authors provide some description of their methods at the bottom of page 7, which could serve better as part of a methods section. I think if the authors intend to present this section as results (alternately, they could restructure so this is not framed as results), this manuscript could benefit from a more explicit, even if brief, section outlining the methodology.
Change made. A new methods section has been added to the paper with information previously provided at the bottom of page 7. In addition, an annex has been included that provides a full listing of national level V&As that have been completed.
P4 line 144: evidence-base, not based
Change made.
P4 lines 132-148: The authors introduce the 1996 work by McMichael et al as laying the groundwork for identifying climate sensitive health impacts. It would be appropriate to refer to some of the more recent works summarizing the literature on climate sensitive health impacts, such as the IPCC reports or more recent work with the WHO.
Change made. Reference to IPCC report and to the WHO quantitative risk assessment report has been added.
P5 line 190: Should this read “decision making needs”?
Change made.
P5 line 204-206: How do the thematic guides described here differ from the V&A guidelines in Canada/PAHO/etc described in the previous paragraph?
Change made. A sentence has been added to indicate how these guides differ from the ones mentioned in the previous paragraph.
P5 lines 204-206: It may be out of scope here, but organizations such as the Climate and Health Alliance in Australia have also been doing work looking at climate change vulnerability and resilience in the healthcare sector.
The authors thank the reviewer for the comment. Given that the focus in the current discussion is on V&A guidelines reference to the work of the Climate and Health Alliance in Australia would indeed seem to be out of scope of the paper.
Table 1: It is not clearly how the detail defined in the Activity column is necessary for the purposes of this manuscript.
The authors thank the reviewer for the comment. As the Discussion and Conclusion sections include information about opportunities and challenges related to conducting V&As that often relate to specific activities and data needs, the authors feel that it is beneficial to keep the more detailed Activity column in Table 1 for the reader’s reference.
P7 lines 232-234: Are there any references to support this statement about resources available for developing country V&A?
Change made. A reference is not available as this estimate was made by WHO staff involved in these assessments, and authors on this paper. The text has been changed to explicitly indicate that these assessments include those supported by WHO.
P7 line 242, P9 line 299: What is the difference between a V&A and a climate change and health assessment as referred to here.
Change made. There is no difference. Use of the term V&A has been made consistent in the text.
P8 line 273-282: Is this all of the assessments identified, or just all of the assessments from the sample of WHO V&As and analyzed in the unpublished WHO report in reference 43? It would be valuable to compare these to V&A that didn’t follow the WHO framework – did those other V&A identify similar health risks, how did they compare in terms of future impact assessment or adaptation planning.
Change made. The text has been revised to indicate that this sentence is referring to the same of WHO V&As. Comparing the V&As analyzed in the unpublished WHO report with others conducted may indeed provide some insightful findings. This is outside of the scope of the current study but would be a useful research question for a follow-up project.
P9 line 301: to prepare [for] and respond to…
Change made.
P9 line 302: Table 2 only refers to US examples (using the BRACE framework), this should be explicit in the text as well, i.e. US states and cities…
Change made.
Have the authors explored including reference to other subnational estimates, such as those in Australia and Canada, or those supported by UN HABITAT? For example, there are references to relevant examples included in the IJERPH special issue.
Change made. Reference to subnational assessments in Canada and Australia have been added.
Table 2: Needs a title.
Change made.
P9 lines 293-304: One of the main findings from BRACE was knowledge and the need for linkages to research institutions to support capacity for local modelling of impacts. The authors should consider describing those here as they refer to them as barriers in the discussion.
Change made. Information on the findings from BRACE regarding the need to build capacity have been integrated into the text.
Discussion
The authors discuss how these V&A are not translating into adaptation practice, however, a major barrier is the lack of evidence informed activities to support adaptation practice. Further, adaptation does not always happen at a national level, it could be that national V&A are leading to adaptation practice at a subnational level which may be more difficult to capture in this type of scoping review.
Change made. This is an excellent point. Text has been revised to provide detail around such barriers. As well, text has been added to indicate that national V&As may be leading to health adaptation at other levels.
P9 lines 312-318: What is the benefit of repeating this list from page 3? If this is a key theme, perhaps it could be highlighted in a table, figure or box and referred to in brief in the text.
Change made. This list was repeated in error. The list on page 3 has been deleted.
P10 lines 325-328: This tool seems important in terms of V&A process, it might fit better in the results section.
Change made. Description of the tool has been moved to the Results section.
P10 line 330: C is missing from Climate in the first line of section 4.1
Change made.
P10 lines 365-369: This point is not highlighted clearly elsewhere. The results discuss the limited resources in developing countries, however, challenges related to data and model availability have not been emphasized previously.
Change made. The finding of the WHO report regarding the number of assessments including either qualitative or quantitative projections of future health impacts of climate change has been added to the previous section.
P10 line 369 – p 11 line 372: Ebi et al in this IJERPH special issue provide a discussion of indicator development for modelling and evaluation of climate related hazards that is relevant here and in the later “way forward” section.
Change made. Key messages regarding the need for robust indicators related to climate change and health impacts and adaptation/resilience have been added to the sections on Assessment Barriers and Way Forward.
P11 lines 392-403: There disconnect between population health and the healthcare sector may act as a barrier to translation of population health research, e.g. into climate sensitive health outcomes and measuring of impacts and producing action plans within health sectors.
This comment is a bit unclear.
P11 lines 420-421: Which WHO report is this referring to?
Change made. An in text citation has been added to identify the WHO report.
P12 line 433: There is a missing period before “As”.
Change made.
P12 lines 446-447: These V&A often do not represent primary research, but rather a synthesis of other published research and so inclusion in the IPCC reports may not necessarily be appropriate.
Change made. Sentence deleted.
P12 line 448: I believe this should be section 4.3
Change made.
P12 line 448: The previous section clearly identifies barriers to V&A and adaptation planning, this section is an opportunity to describe facilitators for completing (and actioning) V&A.
Change made. The list of bullets in this section has been revised and text has been added to describe opportunities through V&As for facilitating and actioning adaptation.
P12 lines 462-463: Should community and other social services be considered as stakeholders here?
Change made. Community and other social services has been added as an example of a stakeholder
P12 line 463: Does “met services” refer to meteorological services?
Change made.
P12 lines 465-466: Indigenous voices and traditional knowledge are an important resource, but this has not been discussed elsewhere in the manuscript. This list seems more a result than a discussion.
Change made. The text has been revised – as per comments above.
P12 lines 467-468: The previous section of the discussion talks in detail about barriers to local assessment. The authors have not provided suggestions on how to resolve this challenge or overcome this barrier. Further, there is limited reference to local (as in subnational) assessments in previous sections.
Change made. Text has been added to the bullet to offer suggestions for obtaining knowledge of climate change impacts on health at the local level. In addition, subnational assessments in Canada and Australia have been added in lines 366-368.
Conclusion
P13 lines 504-506: This point about funding for climate change V&A is important, but it does not seem appropriate to introduce this new information in the conclusion. Consider moving this to section 4.2 where barriers are discussed.
Change made. The discussion about funding has been moved to section 4.2.
References
4: This reference is incomplete.

Reviewer 2 Report
Interesting paper, I enjoyed reading it.
One major concern though is that the methodology part of the manuscript missing. You have not told the reader how you do did what you did. Also related to this, it is not entirely clear to me what you are doing, is this a review of all current V&As to determine/compare progress? Or is it a review of the process of how V&As are developed? Your introduction suggests both? But the process of how you undertook this needs to be fully developed; what literature was reviewed, which databases were used and why was specific literature selected over others? Which countries were studied and why?
There are some very long sentences, which coupled with multiple use of ‘and’ makes reading very difficult. One example from the abstract – ‘ Health authorities and their partners require information about current associations between health outcomes and weather or climate, vulnerable populations, projections of the magnitude and pattern of future risks and adaptation opportunities to reduce exposures, empower individuals to take needed protective actions and build climate resilient health systems.’ There are many others, please re-read entire manuscrit and correct.
Some illogical statements – see line 73 from ‘Knowledge gaps…’, to line 75
References missing? Line 91 to 102 seems to be borrowed from somewhere else?
Author Response
REVIEWER 2
Interesting paper, I enjoyed reading it.
One major concern though is that the methodologypart of the manuscript missing. You have not told the reader how you do did what you did. Also related to this, it is not entirely clear to me what you are doing, is this a review of all current V&As to determine/compare progress? Or is it a review of the process of how V&As are developed? Your introduction suggests both? But the process of how you undertook this needs to be fully developed; what literature was reviewed, which databases were used and why was specific literature selected over others? Which countries were studied and why?
The authors thank the reviewer for the insightful comments on the manuscript. It is hoped that revisions based upon these comments have significantly improved the paper.
A Methods section has been added to the paper to provide this information.
There are some very long sentences, which coupled with multiple use of ‘and’ makes reading very difficult. One example from the abstract – ‘ Health authorities and their partners require information about current associations between health outcomes and weather or climate, vulnerable populations, projections of the magnitude and pattern of future risks and adaptation opportunities to reduce exposures, empower individuals to take needed protective actions and build climate resilient health systems.’ There are many others, please re-read entire manuscrit and correct.
Changes made. A number of sentences have been rewritten to increase clarity.
Some illogical statements – see line 73 from ‘Knowledge gaps…’, to line 75
Change made.
References missing? Line 91 to 102 seems to be borrowed from somewhere else?
Change made. Much of the evidence regarding how climate change has and is altering public health decision making derives from the experiences conducting a number of V&As and the associated engagement with decision makers. The text has been revised to reflect this.

Overall, I felt that this was a nice piece, although it felt a bit repetition and jumbled at times. I got the sense when reading it that different sections had been written by different authors, and insufficient effort had been undertaken to harmonize the sections, resulting in unnecessary repetition and some organizational issues. While the content and research in the piece is solid, I would advise the authors to attempt to edit the piece to reduce its length by reducing repetition and being more succinct in some of the points made. Specific comments are below.
L21—fix this sentence construction “build the global evidence-based of”
L23—when you say “climate change processes” can you be more specific? Are you referring specifically to policymaking processes?
L39—why is “Indigenous” capitalized? This is true throughout the article, and is not a convention I am familiar with.
L41—change “to the impacts” to “to its impacts” or “to the impacts of climate change”
L52/53—can you provide examples of “cascading events, tipping points, and shock events” In my own work, I don’t see these as particularly distinct categories given that many events could be accurately described as overlapping between them, so it would be good to know how you’re thinking of these terms.
L78—It may be worth mentioning around this point that it is not merely extreme weather events associated with climate change that have human health impacts. As you’re well aware, CO2 concentrations by themselves are associated with declines in nutrient availability in staple crops, which has substantial ramifications on public health, particularly in low-income countries where alternatives are limited.
L113—It may be worth replacing “stakeholders” with “policymakers” here
L144—fix this sentence construction “the foundation for the evidence-based on”
L164-169—commas may be more appropriate than semicolons here.
L171—Is there a citation (even just a personal communication) that can back up this claim?
L191—“more thematic and localized information and data” it seems like in addition to this, one of the biggest changes which should be mentioned is the increasing focus of V&As on policymakers as an audience, and structuring V&As to make them more directly applicable to policymaking activities. This should be elaborated on (briefly) more clearly around here.
L237—this sentence needs a citation.
L268—is “project” the same as an “assessment”?
L268—This isn’t strictly necessary, but it would be helpful if possible to briefly describe how the sample you’re describing was generated. Were all of the 34 selected V&As, conducted in LDCs, for instance, or is there some other characteristic indicating why they were chosen?
L299—need “to” between “cities” and “conduct”
L301—need “for” after “prepare”
L304—There is no caption or description for Table 2.
L308-320—This is completely repetitive of what was said earlier in L92-102. I don’t mind summarizing earlier points, but I would dissuade you from repeating the same language twice in the same paper.
L330—“limate” should be “Climate”
L393—Excise “quite specific”
L422—what is meant by the phrasing “past and current diseases”? This needs to be made clearer.
L433—need period after “sector”.
L448—This should be section 4.3 if 4.2 is on barriers.
L502—This topic sentence for the last paragraph seems oddly placed to me. If you want to discuss funding limitations, a perfectly appropriate topic, this text seems to be better placed in Section 4.2 on barriers to conducting V&As. I would replace this paragraph with a broader conclusion about the need for greater engagement in V&As globally.
Author Response
REVIEWER 3
Overall, I felt that this was a nice piece, although it felt a bit repetition and jumbled at times. I got the sense when reading it that different sections had been written by different authors, and insufficient effort had been undertaken to harmonize the sections, resulting in unnecessary repetition and some organizational issues. While the content and research in the piece is solid, I would advise the authors to attempt to edit the piece to reduce its length by reducing repetition and being more succinct in some of the points made. Specific comments are below.
The authors thank the reviewer for the insightful comments on the manuscript. It is hoped that revisions based upon these comments have significantly improved the paper.
Repetition in the paper has been reduced, for example, deletion of lines 94 – 102 on page 3.
L21—fix this sentence construction “build the global evidence-based of”
Change made.L23—when you say “climate change processes” can you be more specific? Are you referring specifically to policymaking processes?
Change made.
L39—why is “Indigenous” capitalized? This is true throughout the article, and is not a convention I am familiar with.
Change made.L41—change “to the impacts” to “to its impacts” or “to the impacts of climate change”
Change made.
L52/53—can you provide examples of “cascading events, tipping points, and shock events” In my own work, I don’t see these as particularly distinct categories given that many events could be accurately described as overlapping between them, so it would be good to know how you’re thinking of these terms.
Change made, “cascading events” has been
deleted and citation has been provided for definition of “shock events” which
may not be familiar to the reader.
L78—It may be worth mentioning around this point that it is not merely extreme weather events associated with climate change that have human health impacts. As you’re well aware, CO2 concentrations by themselves are associated with declines in nutrient availability in staple crops, which has substantial ramifications on public health, particularly in low-income countries where alternatives are limited.
Change made.
L113—It may be worth replacing “stakeholders” with “policymakers” here
The authors thank the reviewer for the
comment. As there are many stakeholders that are not policymakers that use and
rely on information from climate change and health assessments (e.g.
non-governmental organizations, community groups, researchers) the term
stakeholders has been retained in the sentence.
L144—fix this sentence construction “the foundation for the evidence-based on”
Change made.L164-169—commas may be more appropriate than semicolons here.
Change made.L171—Is there a citation (even just a personal communication) that can back up this claim?
Change made. There is no formal reference
for this statement as this was reported to WHO by a number of countries that
used this guidance document, including to WHO authors on this manuscript. The
text has been revised to reflect this.
L191—“more thematic and localized information and data” it seems like in addition to this, one of the biggest changes which should be mentioned is the increasing focus of V&As on policymakers as an audience, and structuring V&As to make them more directly applicable to policymaking activities. This should be elaborated on (briefly) more clearly around here.
Change made. Text has been added that provides an example of how the Ontario Climate Change and Health Toolkit provides important information and guidance that supports health adaptation.L237—this sentence needs a citation.
Change made.L268—is “project” the same as an “assessment”?
Change made.L268—This isn’t strictly necessary, but it would be helpful if possible to briefly describe how the sample you’re describing was generated. Were all of the 34 selected V&As, conducted in LDCs, for instance, or is there some other characteristic indicating why they were chosen?
Change made. Text has been added to describe how the
sample was generated.
L299—need “to” between “cities” and “conduct”
Change made.
L301—need “for” after “prepare”
Change made.L304—There is no caption or description for Table 2.
Change made. A title has been added for
Table 2.
L308-320—This is completely repetitive of what was said earlier in L92-102. I don’t mind summarizing earlier points, but I would dissuade you from repeating the same language twice in the same paper.
Change made. The text has been deleted.
L330—“limate” should be “Climate”
Change made.
L393—Excise “quite specific”
Change made.
L422—what is meant by the phrasing “past and current diseases”? This needs to be made clearer.
Change made to clarify the meaning of the text.L433—need period after “sector”.
Change made.L448—This should be section 4.3 if 4.2 is on barriers.
Change made.
L502—This topic sentence for the last paragraph seems oddly placed to me. If you want to discuss funding limitations, a perfectly appropriate topic, this text seems to be better placed in Section 4.2 on barriers to conducting V&As. I would replace this paragraph with a broader conclusion about the need for greater engagement in V&As globally.
Change made. The discussion about funding
limitations has been moved to the Results section.
